# The Role of Mindfulness in Business Administration (B.A.) University Students’ Career Prospects and Concerns about the Future

**DOI:** 10.3390/ijerph19031376

**Published:** 2022-01-26

**Authors:** Manel Plana-Farran, Àngel Blanch, Silvia Solé

**Affiliations:** 1Department of Business Administration, University of Lleida, 25001 Lleida, Spain; 2Department of Psychology, University of Lleida, 25001 Lleida, Spain; angel.blanch@udl.cat; 3Faculty of Nursery and Physiotherapy, University of Lleida, 25198 Lleida, Spain; silvia.sole@udl.cat; 4Institute of Biomedical Research (IRB Lleida), 25198 Lleida, Spain

**Keywords:** mindfulness, concerns, entrepreneurship orientation, management

## Abstract

In a challenging work environment, entrepreneurship orientation (hereafter, EO) can be an important asset for university students. In this study, we investigated the EO and concerns about the future of B.A. students, focusing on the role of mindfulness levels. A total of 204 students, including those coming from family businesses (hereafter, FB), were asked about their intention of creating their own business and future concerns with an ad hoc questionnaire, and about their mindfulness levels with the Mindful Attention Awareness Scale. The results showed no differences in concerns about the future between those students coming from a family business and those who did not. However, in that group of students who were uncertain about starting a business career, a negative association between mindfulness and future concerns was found. This paper sheds light on the relationship between mindfulness and concerns about the future in Business Administration (hereafter, B.A.) students, concluding that, in the face of uncertainty, higher levels of mindfulness help to reduce concerns about the future. Finally, we indicate the relevance of this study for entrepreneurs, family business members, policymakers and B.A. Faculties and Business schools.

## 1. Introduction

In the changing prevailing and contemporary environment, the willingness to initiate a work career is a key element concerning occupational commitment [1]. New employment opportunities relate to entrepreneurship [2]. In a dramatically changing work and economic environment, several skills needed for career development are receiving increased attention from researchers worldwide [3,4].

There is a stream of research about the use of mindfulness in considerable scientific fields including psychology, neuroscience, medicine, education, business management, and entrepreneurship [5,6,7,8,9]. Mindfulness is defined as “a receptive attention to and awareness of present events and experience” [5] (p. 212). In addition to this, the effects that mindfulness plays in organizational and business settings have been considered using samples of employees and principals [6,7,10,11,12,13,14], while the topic has been short explored in entrepreneurship [9] and in Family Business [15,16]. In this vein, Van Gelderen et al. [9] (p. 490) state that “mindfulness is a relevant concept in the context of entrepreneurship” and Arzubiaga et al. [16] sustain that mindfulness levels in potential successors’ of family business offer relevant and insightful managerial implications for firm’s future.

Family enterprises represent singular organizations that need to manage several emotional logics to survive in the long term [17]. Those university students belonging to a family business bear a considerable emotional attachment [18]. On the other hand, uncertainty is quite common in university students [19]; hence, mindfulness has emerged as an important concept because it stimulates the capacity to focus on the present [20]. However, whether mindfulness relates to the personal and professional concerns of close-to-graduation business students remains relatively unknown. This association could be moderated by factors such as belonging to a family already running a business, or to future education or self-employment prospects.

The role of mindfulness in FB and Entrepreneurship has attracted considerable attention, even though it remains relatively unexplored concerning its impact on B.A. students’ career prospects, and concerns about the future. To fill this gap and to complement previous studies, we contribute to research showing mindfulness as an effective tool in regulating emotions among business students, appearing as a particularly useful protective factor when coping with uncertainty. 

In the current study, we address this topic with a sample of undergraduate students enrolled in business and economics studies. More specifically, we evaluate whether the association of mindfulness with professional and educational prospects depends on whether one belongs to a family running a business or not, to one’s intentions to engage in further education, or to one’s intentions to become self-employed. The aim of this study is to evaluate whether mindfulness is effective in regulating emotions among B.A. students related to professional and personal concerns.

We mainly obtained that there is a negative association of mindfulness with future professional and personal concerns with greater uncertainty about becoming self-employed. In consequence higher mindfulness levels implied lower future levels of concern in the group of students who were uncertain about creating their own business. Mindfulness appears effective in regulating emotions among business students to cope with uncertainty about their self-employment prospects. 

### 1.1. Family Business (FB)

A FB is defined mainly by the fact of being owned and managed by family members [21]. That means the possibility of a partnership between spouses or siblings and, importantly, across generations [22]. In this vein, Berrone et al., and Miller et al. [23,24] maintain that FBs are characterized by visible and active owners, a long-term orientation, a collective identity, and a desire for the firm to persist across generations. Furthermore, there is a considerable consensus regarding the core elements characterizing a family business. Thus, authors such as Lansberg et al. [25] establish property as a basic element, whereby the members of the family bear the legal control. In addition, the continuity and existence of generational change provide a basic and fundamental distinction for something to be considered a family business. An increasing number of studies add the self-perception of belonging to a family business [26,27].

According to Cabrera and Santana [28], the differences between an FB and a non-family business (nonFB) are based on four aspects: structural, operational/functional, managerial, and strategic, which in turn lead to a set of strengths and weaknesses. Thus, Barroso [26] establishes that within the weaknesses there are succession and emotional conflicts. Kenyon-Rouvinez et al. [29] assume that more conflicts tend to arise in FBs than in nonFBs due to the overlap between the family system, management, and ownership.

The intensive intertwined family and business subsystems in FBs provoke an intense interaction between both systems [30]. Family firms have been recognized as complex entities where family dimensions and business and commercial logic need to be fit in a single organization [31]. The family and the business subsystems have been represented as two not-aligned social systems that represent emotional and business scenarios, respectively [16,32].

### 1.2. Entrepreneurial Orientation (EO), Entrepreneurial Intention (EI) and Concerns about the Future

The individual-level phenomenon of entrepreneurship [33] relies, mainly, on the question “How does an entrepreneur create a venture?” [34]. Entrepreneurship and the importance of understanding entrepreneurs and their mental processes is not only a key topic [35,36] but an essential aspect to broad knowledge related to the patterns of behaviour of entrepreneurs [37]. 

EO is a crucial driver of entrepreneurial activities in business performance at the individual level [38], and an important element of EI [39]. This latter concept differentiates entrepreneurs from non-entrepreneurs depending on risk-taking, pro-activity, and the ability to capture a business opportunity. EI is the behavior in the business aspect, whereas EO focuses on individuals as organizations [40].

McCline et al. [41] measure EO throughout the degree of innovation in opportunity, achievement, and self-esteem, distinguishing between entrepreneurs and non-entrepreneurs. On the other hand, authors such as Gaglio and Katz [42] do not perceive significant differences between entrepreneurs and non-entrepreneurs. Risk-taking management guides individuals toward action, which is known as the willingness to perform an activity with unknown outcomes [43]. EO means an action-behavior that motivates one to venture into the unknown in the interest of obtaining a self-occupation activity or returns by grasping opportunities in the marketplace. Gorostiaga et al. [44] define as part of EO the assumption of risks associated with decisions with a marked orientation toward achievement and learning. According to Craig et al. [45], Arzubiaga et al. [46], and Pressuti et al. [47], EO is characterized by risk-taking, innovativeness, proactiveness, and the need for searching for different opportunities [48]. Keh et al. [49] maintain there is a fundamental difference in the thinking processes of entrepreneurs when they evaluate and analyze opportunities.

Young people studying management and business administration are expected to be future companies’ builders and leaders. They are supposed to become entrepreneurs, but on many occasions, they have not been prepared to enhance their “entrepreneurship capacity” [44]. There is a large scholarship based on the behavior of entrepreneurs enrolled in the process of producing new ventures [50]. For instance, Kusa et al. [48], Olugbola [51], and Edelman et al. [52] explored the role of entrepreneur motivation along with its impact on the decision to start a company and the results obtained. In contrast, there is a vague standpoint that referred to the view of B.A. students in the previous phase of creating and running a business, specifically on uncertainty and professional concerns. 

### 1.3. The Role of Mindfulness in Emotions and Concerns in University Field

Many studies have addressed the stress levels of university students caused by factors such as the pressure to achieve certain qualifications or continuous working pressure [53,54]. In addition, a good adaptation to the university environment is needed, and uncertainty about the future is always present [19,55].

Mindfulness has become an important concept when talking about mental health. It is defined as focusing on the present moment in a non-judgmental and accepting way [56]. It can be a dispositional trait or a state that can be trained with several studies showing its efficacy in improving mental health and diminishing stress and anxiety in different populations [20,57,58]. 

Regarding university students, mindfulness can reduce anxiety, depression, and academic stress, even having a protecting role for burnout in healthcare students [59,60]. In this vein, mindfulness-based educational interventions have shown the improvement of the social climate in university classrooms [61]. Furthermore, academic performance is enhanced in those students that practice mindfulness meditation regularly [62]. Mindfulness and student performance has been confirmed among university students in five countries [63]. Finally, mindfulness training has proved as a source of improving attention in university classes [64]. 

Nonetheless, to our knowledge, there are no mindfulness interventions in management university students, but the benefits of mindfulness in business organizations are being increasingly used. There are researchers who differentiate between mindfulness on individual and collective levels, defining this last as a sort of psychological state of a team or an organization [65,66]. Previous research about mindfulness in business shows seven processes that benefit from it: attention, awareness, cognition, self-regulation of behavior, emotions, thoughts, and physiology [7].

As a consequence of these factors, mindfulness has the capacity to help people to focus on the present instead of being anchored in the past or anticipating the future [20] so it can be expected that it has a protecting role in the face of uncertainty and future concerns.

### 1.4. The Present Study

University students are continuously facing different decisions about the future such as the continuation of studies or the possibility of creating their own business. The willingness to create a new business relates to entrepreneurial orientation, which is a key point when talking about the students of management degrees. In the case of students who are members of a family with a business, they also must manage this increased pressure [67].

Mindfulness is a new but well-consolidated concept that relates to the capacity to be aware of the present moment and it has been demonstrated to be useful for helping people to reduce their stress and anxiety about the future.

This study addressed the association of mindfulness with the personal and professional concerns of undergraduate students pursuing a career in business administration. This group is particularly suited to address this topic because these students may be prone to undertake the building of their own business or continue with the family business in case they belong to one. More precisely, we examined whether the association between mindfulness with personal and professional concerns varied depending on whether the participants belonged to a family business, and whether they were certain or uncertain about pursuing further professional training or going into self-employment.

## 2. Method

### 2.1. Participants and Context

The participants were 204 university students enrolled in degrees in Business Administration as well as Business Administration and Law at the University of Lleida (UdL). According to Zellwegger [32], the dataset is based on the responses of students who have not started professional careers yet but who are training and preparing for them. The formal program, consisting of lectures and seminar groups, was the same for all the participating students. Of the total of 204, 11 students had to be excluded from the study because they did not answer the questions or failed to answer some of the questions in the questionnaire survey developed by Gross [68]. Thus, the final sample consisted of 193 students. The statistical power for this sample size to detect a 0.2 correlation at the 0.05 alpha level is around 80% [69] (R Core Team, 2019). Participants’ ages ranged from 17 to 30 years old, with a median of 19. The mean age was 20.1 years, with a standard deviation of 2.38.

### 2.2. Measures

A questionnaire was created ad hoc for this study. Students were asked about self-employment, personal and professional concerns about the future, which degree they were studying, and if they were coming from a family business or not.

Professional and personal future concerns: the students were asked if they were worried about their personal and professional future. The answers could be “yes”, “no”, or “I don’t know”. Self-employment and education prospects: the students were asked about their intention to keep on studying and creating their own business, and the answers could be “yes”, “no”, or “I don’t know”. Mindfulness was measured with a six-item version of the MAAS-Mindfulness Attention Awareness Scale [70]. This shorter scale yielded a Cronbach’s alpha reliability coefficient of 0.80.

### 2.3. Procedure

The sample of students was obtained from a University Business Conference held in the city of Lleida in 2018. Participation was voluntary and anonymous. A brief ten-minute explanation was offered to the students to prevent any potential misunderstandings related to the questionnaire. At the end of the conference, those individuals who wanted to participate were administered the two questionnaires, which they answered on site.

### 2.4. Statistical Analyses

Figure 1 shows a structural equation model with two latent variables, Mindfulness and Concerns. Mindfulness was measured with the six items comprising the scale used. The concerns variable was measured with two observed indicators of professional and personal concerns about the future. This model was contrasted across the three moderators: a family running or not running a business, educational prospects, and self-employment prospects.

Two models were contrasted between the two levels of each moderator. The first model constrained the path linking Mindfulness and Concerns to be equal across the two levels of each moderator. The second model freed this path to vary across the two levels of the moderator. Both models were contrasted with a chi-square difference test [71], with a significant difference test being supportive of meaningful differences across the respective levels of the evaluated moderators.

Additional fit indices to evaluate the estimated models were the comparative fit index (CFI), the Tucker–Lewis Index (TLI), the root mean square error of approximation (RMSEA), and the Akaike’s Information Criteria (AIC), with CFI and TLI values close to one, RMSEA < 0.09, and lower AIC values being supportive of a good fit to the observed data. The data analyses were conducted with the lavaan package from the software R [69,72]. The data and code used to implement the current statistical analyses are available from the corresponding author.

## 3. Results

Table 1 shows the descriptive statistics of mindfulness and professional and personal concerns about the future across the three studied moderators, family business, educational prospects, and self-employment prospects. Differences in the three variables were minimal across the moderator’s levels. Only medium effect sizes were found for professional concerns regarding family business (*d* = 0.28), and mindfulness regarding self-employment prospects (*d* = 0.26).

Table 2 shows the parameter estimates of the analyzed structural equation model (Figure 1). For the family business and educational prospects moderators, the association of mindfulness with concerns about the future was very similar at both levels of both moderators. The model fit was acceptable for either model, even though the chi-square difference tests across equal and different parameter models were non-significant for family business (Δχ^2^ = 0.11, *p* = 0.917) and education prospects (Δχ^2^ = 0.12, *p* = 0.733).There was only for self-employment prospects a statistically significant negative association of mindfulness with concerns (−0.37, *p* < 0.05) for those individuals who were uncertain about entering self-employment, whereas it was virtually null for those individuals wishing to set up their own business.

This effect was weak, with individual differences in mindfulness only explaining 14% of the variability in professional and personal concerns about the future, even though there was an acceptable fit to the observed data (CFI = 0.961, TLI = 0.942, RMSEA = 0.058, AIC = 4013). Moreover, the chi-square difference test across the equal and different parameters of the model was only close to significance (Δχ^2^ = 2.54, *p* = 0.111). These findings suggest, however, that mindfulness might somewhat buffer the psychological negative effects of the dilemma of what path to undertake when finishing university training.

## 4. Discussion

The present study addressed whether mindfulness was associated with prospective personal and professional concerns with a sample of undergraduate students. These associations were examined regarding three moderators, the presence of a family business, further education intentions, and self-employment intentions [73].

The main findings suggest that mindfulness did not influence future professional and personal concerns regardless of whether individuals belonged to a family business or whether they were willing to pursue further education. These results agree with recent findings described by Arzubiaga et al. [16], where no differences in mindfulness levels emerged between members and non-members of family businesses. Although some findings argue about the theoretical effects of mindfulness on family business [32], there is a paucity of research measuring mindfulness levels on real family businesses.

There was a weak negative association of mindfulness with future professional and personal concerns with greater uncertainty about becoming eventually self-employed. This means that higher mindfulness levels implied lower future levels of concern in the group of students who were uncertain about creating their own business. Therefore, mindfulness might be effective in regulating emotions among business students. In addition, mindfulness might promote buffering effects between those who have uncertainty about their self-employment prospects because of facilitating more adaptability and open-minded thinking and enhancing EI, learning skills, and education and self-employment prospects. Furthermore, mindfulness may be beneficial to students’ emotional attachment as it generates a low level of worries related to business students. These results are in line with that literature that shows how mindfulness can help us to focus on the present moment instead of being anchored in the past or thinking about the future and even creating fearful expectations [20]. Notably, the roles of emotion regulation and mindfulness emphasize the advantages of embodying mindfulness practices in business administration students.

### Limitations and Future Directions

This study has some limitations to bear in mind. The sample is of limited size and the conclusions of the study are likely to be constrained to students pursuing careers in business. This was the first pilot study in the context of a single and specific subject in a B.A. Faculty. Future research should consider larger sample sizes to confirm or refute these preliminary results. Moreover, this study focuses on a single university from a southern European country. For this reason, the results may not be representative enough to explain the B.A. students’ behavior and their mindfulness levels when facing the pressures of uncertainty. On the other hand, the statistical effect was weak, that is another limitation, but the strength of this study is that the results are a very new finding in the field of management and career development. This study can open a new line of research when talking about ways to help undergraduate students manage their uncertainty regarding the future.

In addition, we suggest that B.A. studies, programs, and researchers expand investigation on B.A. students’ emotional experiences, membership of FBs, and emotion regulation to enhance their capacity to aim for business challenges. Based on the present results, the university B.A. education system would benefit from the development or incorporation of a training program for students’ mindfulness regulation. Through the emotion regulation program, B.A. students would be able to first recognize their emotions, and then better understand the emotions of their labor and business environment. It would be beneficial to develop a program or at least to embody a subject in business degree. We propose integrating classes related to mindfulness and emotion into the B.A. curriculum, providing the opportunity to acquire and practice mindfulness skills, which are required to facilitate better management and improvement of the quality of life.

## 5. Conclusions

Mindfulness may be effective in regulating emotions among businesses students in the face of business-degree uncertainty related to labor and self-employment prospects. Moreover, the condition of a being member of an FB or not, does not affect mindfulness levels in the same vein of previous studies [16]. Regarding prospect B.A. students’ careers, higher mindfulness levels implied lower future levels of concern in the group of students who did not know whether they wanted to create their own business. To our knowledge, this is the first study analyzing the role of mindfulness in the concerns about the future of B.A. students’, and it extends the existing research regarding this subject.

Mindfulness may be effective in regulating emotions among business students as it promotes a buffering effect for those who have uncertainty related to their self-employment prospects. Although these preliminary results must be confirmed in further studies with bigger samples, it seems that mindfulness can result in a protector element for the mental health of B.A. students, and new research along these lines is warranted in the future.

Our research also brings forward some implications for B.A. studies and curricula. First, it emphasizes the concern for emotional regulation for B.A. students. This goes in line with a global development priority included in United Nations Sustainable Development Goals (SDGs) as mental health and psychosocial wellbeing [74]. Hence, University B.A. programs may consider the specific nature and challenges related to EO and EI context. We add to the body of practical implications the suggestion of integrating classes or practices related to mindfulness in B.A. studies and programs or to embody a subject in business degree providing the knowledge and the practice of mindfulness as a source of regulating emotions.

Our broader scope of emotional regulation and mental health in B.A. is aligned with the paradigm shift in the field of psychology related to entrepreneurship [34,48,75] and family business [30]. In consequence, it might be worth for policymakers to bear in mind a requirement for introducing a mindfulness program in B.A. studies in line with Asthlana [76] that found mindfulness as a highly significant predictor of increasing the marks obtained by MBA students.

## Figures and Tables

**Figure 1 ijerph-19-01376-f001:**
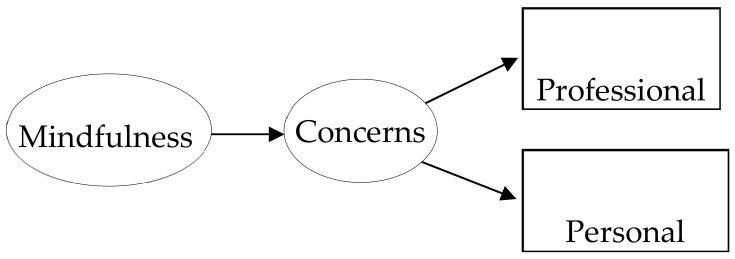
Structural equation model relating mindfulness with professional and personal concerns about the future. This model was contrasted across belonging or not to a family that was running a business, and across whether there were educational and self-employment prospects or not. The mindfulness latent variable was measured with six items (7, 8, 9, 10, 13, and 14) from the classical Mindfulness Attention Awareness Scale.

**Table 1 ijerph-19-01376-t001:** Means (M), standard deviations (Sd), and effect sizes (*d*) of mindfulness, and professional and personal concerns about the future across belonging to a family business (Yes, No), and education and self-employment prospects (Yes, Uncertain).

	Family Business			Education Prospects			Self-Employment Prospects		
	Yes	No		Yes	Uncertain		Yes	Uncertain	
Variable	M (Sd)	M (Sd)	*d*	M (Sd)	M (Sd)	*d*	M (Sd)	M (Sd)	*d*
Mindfulness	24.7 (5.5)	24.9 (4.8)	0.04	24.8 (4.6)	24.8 (5.8)	0.00	24.22 (4.9)	25.5 (4.6)	0.26
Professional concerns	3.5 (0.8)	3.8 (0.9)	0.28	3.7 (0.92)	3.6 (0.74)	0.06	3.66 (0.91)	3.67 (0.79)	0.02
Personal concerns	3.3 (1.1)	3.3 (1.1)	0.05	3.3 (1.2)	3.3 (0.99)	0.03	3.33 (1.15)	3.24 (1.03)	0.08

**Table 2 ijerph-19-01376-t002:** Parameter estimates and fit indices showing the association of mindfulness with future concerns (professional and personal) by family business, education prospects, and self-employment prospects.

	Family Business		Education Prospects		Self-Employment Prospects	
	Yes	No	Yes	Uncertain	Yes	Uncertain
Mindfulness → Concerns	−0.13	−0.11	−0.08	−0.16	0.02	−0.37 *
*R* ^2^	0.02	0.01	0.01	0.03	0.00	0.14
χ^2^	48.82		59.01		48.16	
df	39		39		38	
CFI	0.968		0.935		0.961	
TLI	0.954		0.906		0.942	
RMSEA	0.051		0.080		0.058	
AIC	4463		4313		4013	
Δ*χ*^2^	0.11		0.12		2.54	

Note. * *p* < 0.05.

## Data Availability

The data wil be made available on request from the corresponding author.

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
