# Peer review of "The Role of Mindfulness in Business Administration (B.A.) University Students’ Career Prospects and Concerns about the Future"

_ijerph, 2022, doi:10.3390/ijerph19031376_

Round 1
Reviewer 1 Report
This is an interesting paper and I enjoyed reading it. However, there are essential weaknesses that need to be addressed.
0) Abstract: Authors should state their contribution in terms of issue problems solved or ameliorated, theory or policy dilemmas resolved, or the like. Abstract should offer at least one example of a theoretical or managerial implication that authors concluded after their work.
1) The introductory/opening section should communicate a little clearer the literature gaps, as well as the study's aims & objectives in order to facilitate the flow of the study.
2) Additional references to recent & relevant empirical studies could increase the quality of the research paper and provide a much clearer message to the reader - these may help you building your discussion which needs to be extended. Add the following to your reference list:
Kusa, R., Duda, J., & Suder, M. (2021). Explaining SME performance with fsQCA: The role of entrepreneurial orientation, entrepreneur motivation, and opportunity perception. Journal of Innovation & Knowledge. 6(4), 234-245.
Metallo, C., Agrifoglio, R., Briganti, P., Mercurio, L., & Ferrara, M. (2021). Entrepreneurial Behaviour and New Venture Creation: the Psychoanalytic Perspective. Journal of Innovation & Knowledge, 6(1), 35-42.
Piñeiro-Chousa, J., López-Cabarcos, M. Á., Caby, J., & Šević, A. (2021). The influence of investor sentiment on the green bond market. Technological Forecasting and Social Change, 162, 120351.
3) The statistical treatment is acceptable.
4) Concluding remarks – authors must elaborate more on what is their contribution to the literature as well as on opportunities for future research. Questions that need to be answered: Why your study is important? and how it extend so existing knowledge on the issue/topic? Conclusions need to be written in a clear and coherent manner and draw the main lessons from the paper. I suggest you to concentrate on the description of the implications of the work, the main findings and its potential replicability - empirical investigation. Furthermore, limitations of the study need to be outlined to a greater extent, and so are any potential connections between your study and specific aspects of the Journal's scope.
5) Carefully check the references, so as to make sure they are all complete and follow the Guidelines to Authors.
6) The paper needs to be revised by an English native speaker. Some expressions need to be revised and given a fresh approach by an experienced native proof reader.
Thank you for the opportunity to read the paper.
Author Response
Dear Reviewer,
Thank you for recognizing the potential of the paper. In the revision process, we paid special attention in the feedback provided by you. We hope you agree that the manuscript has benefitted from this revision. Please see the attachment.
Thanks in advance.

Reviewer 2 Report
Thank you very much for giving me this opportunity to review this article. I found interesting and significant work. 1. Research objectives and significance is missing in the introduction part. Please add these two part. 2. The literature review is too old and not enough. section 1.2 and 1.3 need to add more strong literature . 3. why data were collected from university students and why choose just one university (University of Lleida (UdL).? please provide the answer of these importnat questions in the methods section. 4. The sample size is too small. Could you please justify ? which sampling method was used for data collection. 4. The implications are missing. 6. Results are not enough. Please use some additional variables. Good LcukAuthor Response
Dear Reviewer,
We would like to thank you the recognition of the potential of our work. In the revision process, we have closely followed the suggestions in the feedback provided by you.
Thanks in advance

Round 2
Reviewer 1 Report
.
Reviewer 2 Report
Thank you for revision.